# High-resolution spectroscopy of single nuclear spins via sequential weak measurements

Matthias Pfender[1], Ping Wang[2,3], Hitoshi Sumiya[4], Shinobu Onoda [5], Wen Yang [2], Durga Bhaktavatsala Rao Dasari[1,6], Philipp Neumann[1], Xin-Yu Pan[7], Junichi Isoya [8], Ren-Bao Liu[3] & Jörg Wrachtrup[1,6]

Nuclear magnetic resonance (NMR) of single spins have recently been detected by quantum sensors. However, the spectral resolution has been limited by the sensor's relaxation to a few kHz at room temperature. This can be improved by using quantum memories, at the expense of sensitivity. In contrast, classical signals can be measured with exceptional spectral resolution by using continuous measurement techniques, without compromising sensitivity. When applied to single-spin NMR, it is critical to overcome the impact of back action inherent of quantum measurement. Here we report sequential weak measurements on a single $^{13}$C nuclear spin. The back-action causes the spin to undergo a quantum dynamics phase transition from coherent trapping to coherent oscillation. Single-spin NMR at room-temperature with a spectral resolution of 3.8 Hz is achieved. These results enable the use of measurement-correlation schemes for the detection of very weakly coupled single spins.

[1] 3rd Institute of Physics, Research Center SCoPE and IQST, University of Stuttgart, 70569 Stuttgart, Germany. [2] Beijing Computational Science Research Center, Beijing 100193, China. [3] Department of Physics & Centre for Quantum Coherence, The Chinese University of Hong Kong, Shatin, New Territories, China. [4] Sumitomo Electric Industries Ltd., Itami 664-0016, Japan. [5] Takasaki Advanced Radiation Research Institute, National Institutes for Quantum and Radiological Science and Technology, Takasaki 370-1292, Japan. [6] Max Planck Institute for Solid State Research, Stuttgart 70569, Germany. [7] Institute of Physics, Chinese Academy of Sciences, Beijing 100190, China. [8] Faculty of Pure and Applied Sciences, University of Tsukuba, Tsukuba 305-8573, Japan. These authors contributed equally: Matthias Pfender, Ping Wang. Correspondence and requests for materials should be addressed to R.-B.L. (email: rbliu@cuhk.edu.hk) or to J.W. (email: wrachtrup@physik.uni-stuttgart.de)

A quantum measurement induces back-action on the measured system, causing the system to collapse into different states determined by the random measurement outcome. When monitoring the dynamics of a quantum object by sequential measurements, the back-action induces inevitable disturbance. Weak measurements, as introduced theoretically[1–7] and demonstrated experimentally with nitrogen-vacancy (NV) centers[8,9], superconducting qubits[10–12] and other systems[13,14], are a potential solution to approach the limit of negligible disturbance of the system under study. However, this comes at the price of less information on the system.

NV centers in diamond have recently been shown to be exceptional sensors for nanoscale nuclear magnetic resonance (NMR)[15–22]. In these experiments, the NV center typically probes nuclear sample spins in a Ramsey-type experiment, i.e., a free evolution of the spins in between the preparation and readout steps. High-spectral resolution, necessary for chemical identification of molecules, requires long free evolution times. To avoid measurement back-action on the sample spins, these schemes usually avoid measurements during free evolution of the system[18,23–30]. On the other hand, long free evolution times leave the sensor idle and hence decrease sensitivity.

Here we show that, by interleaving sequential measurements and free evolution of the system, back-action can be mitigated while at the same time sensitivity can be increased drastically. In addition, if the measurement rate is larger than the hyperfine coupling to nuclear spins, an effective decoupling of the nuclear spin precession from electron spin decay can be achieved. Exploiting these features of sequential weak measurements, we devise theoretically and demonstrate experimentally a scheme to reconstruct the time evolution of a single nuclear spin from the random results of many subsequent measurements by carefully tuning the strength and timing of measurements. By analyzing the correlations in the measurement record we reach the limit where measurements on single quantum systems are nearly back-action free and reconstruct their dynamics far beyond the limits set by the sensor lifetime.

## Results

**Correlation signal of sequential weak measurements.** Weak measurements are implemented as shown in Fig. 1a. The sensor spin is initially polarized along the $x$-axis, denoted as $|x\rangle$. Essentially, the measurement of the nuclear spin consists of a rotation of the electron spin conditional on the quantum state of the nuclear spin (controlled phase gate). The controlled phase gate $e^{2i\alpha\hat{S}_z\hat{I}_x}$ is based on the magnetic dipole interaction between the sensor spin $\hat{\mathbf{S}}$ and the target spin $\hat{\mathbf{I}}$, with interaction strength $\alpha$ (see Methods). As shown in Fig. 1b, the controlled phase gate causes the sensor spin to rotate about the $z$-axis by an angle $\pm\alpha$ for the target spin state $|\pm X\rangle$ (i.e., polarized parallel or anti-parallel to the $X$-axis). The evolution of the sensor spin for a certain initial state of the target is $|x\rangle \otimes (a|+X\rangle + b|-X\rangle) \rightarrow a|+\alpha\rangle \otimes |+X\rangle + b|-\alpha\rangle \oplus |-X\rangle$ with $|\pm\alpha\rangle$ denoting the sensor state rotated away from the $x$-axis by $\pm\alpha$. The sensor spin is then measured along the $y$-axis. The probabilities of the two outcomes, $m_k = \pm 1$ depend on the initial target state $|\pm X\rangle$. The projective measurement of the sensor therefore constitutes a measurement of the target spin $\mathbf{I}$, with a strength depending on the value of $\alpha$. In particular, a projective (strong) measurement is found for $\alpha = \frac{\pi}{2}$. Weak measurements can also be used for heralded initialization of the target spin along the $X$-axis upon post-selecting one of the outcomes of the sensor measurement[9].

In between two successive measurements, the target spin undergoes free precession about the $Z$-axis by an angle $\Phi$. Thus, the outcomes of any two measurements correlate depending on

the spin precession (see Methods). In the limit $\alpha \to 0$, the measurement-induced dephasing (disturbance) is negligible, and the correlation function between two measurement outcomes separated by $N$ measurements, $C(N) \equiv \langle m_{k+N} m_k \rangle = \sin^2\alpha \cos(N\Phi)$, oscillates at the precession frequency $\Phi$ (in units of radian per measurement cycle) (see Methods), without measurement-induced damping. In this limit, i.e., for very weak measurements the target spin precession frequency can be determined with arbitrary spectral resolution. In practice, however, each measurement dephases the target spin (in the $X$ basis) and the spin dynamics shows a damped Rabi oscillation (about the $Z$-axis). By taking this finite dephasing into account, we find the expression for the correlation function

$$C(N) = \frac{\sin^2\alpha \left(C_+ \eta_+^N + C_- \eta_-^N\right)}{2}, \quad (1)$$

where $C_\pm = 1 \pm \mu \frac{\cos\Phi}{\sqrt{\mu^2 - \sin^2\Phi}}$, $\eta_\pm = \left(\cos\Phi \pm \sqrt{\mu^2 - \sin^2\Phi}\right)\cos^2\frac{\alpha}{2}$ with $\mu \equiv \tan^2\left(\frac{\alpha}{2}\right)$ (which can be regarded as the measurement strength for $\alpha \leq \frac{\pi}{2}$).

There is a phase transition in the quantum dynamics between coherent oscillation and coherent trapping at the boundary $\mu^2 = \sin^2\Phi$, due to the competition between the free precession and measurement-induced dephasing (similar to damped Rabi oscillations)[2,6,7,31,32].

On one hand, when the measurement is relatively weak, i.e., $\mu^2 < \sin^2\Phi$, the correlation function is an oscillatory function with an effective decay of $\gamma_{\text{eff}} = -\frac{1}{2}\ln(\cos\alpha)$ per measurement cycle. This is half the measurement-induced dephasing rate as the measurement dephases the spin only along the $Y$-axis, while the spin is rotating in the $X$–$Y$ plane. Taking this into account, the renormalized angular frequency of the target spin precession is given by (in units of radian per measurement cycle)

$$\Phi_{\text{eff}} = \arccos\frac{\cos\Phi}{\sqrt{1 - \mu^2}}. \quad (2)$$

Due to this, the frequency is either dragged towards 0 (oscillation slowed down) or $\pi$ (oscillation sped up) depending on the precession angle $\Phi$ being less or greater than $\frac{\pi}{2}$ (see Fig. 1c).

On the other hand, when the measurement is relatively strong, i.e., $\mu^2 > \sin^2\Phi$, the correlation function exhibits an exponential decay i.e., $C(N) \sim \exp(-N\gamma_{\text{eff}})$ if $\cos\Phi > 0$ or decay with alternating sign, $C(N) \sim (-1)^N \exp(-N\gamma_{\text{eff}})$ if $\cos\Phi < 0$, with an effective decay rate $\gamma_{\text{eff}} = \min(-\ln|\eta_+|, -\ln|\eta_-|) \approx \sin^2\Phi/[2\tan^2(\alpha/2)]$. This indicates that the spin is trapped approximately along the $X$-axis. The physical picture of the trapped dynamics is illustrated in Fig. 1d. The trapping dynamics is similar to the quantum Zeno effect[31,32], where the trapped states are coherent superpositions of the energy eigenstates $|\pm Z\rangle$—the eigenstates of the free precession.

**Experimental realization of sequential weak measurements.** In our experiments, we use the electron spin of a single NV center in diamond as the quantum probe for single $^{13}$C nuclear spins in close proximity. Experiments are carried out at room temperature, where the spin lifetime of the NV center spin is on the order of milliseconds. The diamond crystal in use has a $^{13}$C abundance depleted to 0.005%. The electron spin $\hat{\mathbf{S}}$ and the $^{13}$C nuclear spin $\hat{\mathbf{I}}$ are coupled via the hyperfine interaction $\hat{S}_z\mathbf{A}\cdot\hat{\mathbf{I}}$, where the $z$-axis is along the NV symmetry axis and the hyperfine interaction strength $A \sim$ kHz. We apply an external magnetic field $B \approx 2,561$ Gs along the $z$-axis, and choose the transition between $|m_S = 0\rangle$ and $|m_S = -1\rangle$ as the sensor qubit.

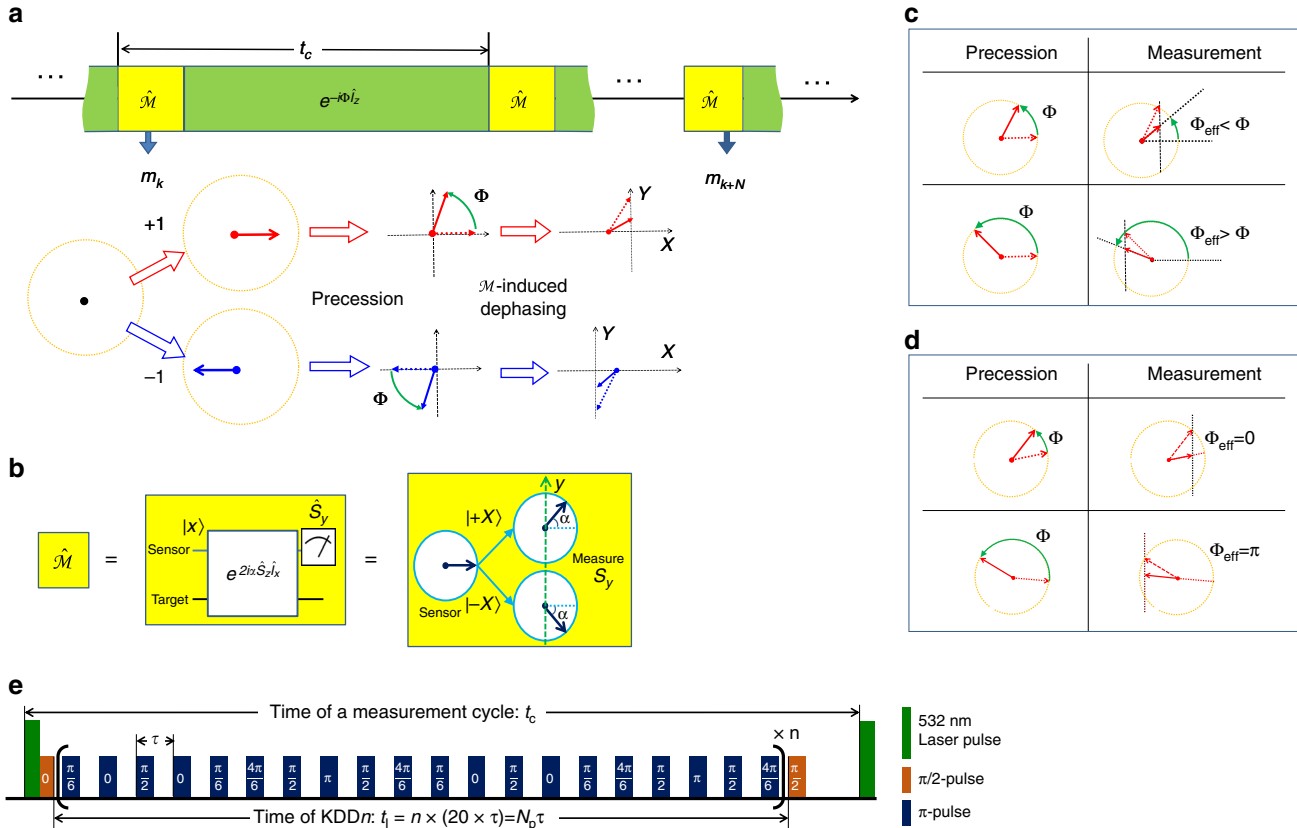

**Fig. 1** Sequential weak measurement of a target spin via projective measurement of a sensor spin. **a** The target spin is partially polarized along $X$ or $-X$ depending on the $k$-th measurement result of the sensor spin, then undergoes free precession about the $Z$-axis. If the measurement result is discarded, the target spin experiences measurement-induced dephasing along the $X$-axis. The correlation between different measurements on the sensor, e.g., the $k$-th and the $(k+N)$-th ones, reflects the spin dynamics. **b** The sensor-assisted weak measurement ($\hat{\mathcal{M}}$) of the target spin. The sensor is initialized along the $x$-axis. By the control-phase gate, the sensor spin precesses about the $z$-axis by opposite angles for opposite target states $|\pm X\rangle$. A projective measurement on the sensor along the $y$-axis constitutes a weak measurement of the target spin. **c, d** Precession by an angle $\Phi$ (left column) and the measurement-induced dephasing (right column), shown as evolution from dotted arrows to solid ones. **c** The effective precession $\Phi_{\text{eff}}$ is slowed down/sped up for $\Phi$ greater/less than $\frac{\pi}{2}$. **d** For $\Phi$ close to 0 or $\pi$, the spin is coherently trapped along an axis ($\Phi_{\text{eff}} = 0$) or alternatively along opposite directions ($\Phi_{\text{eff}} = \pi$). **e** In the experiment, a measurement cycle contains a 532 nm laser pulse (300 ns), a KDD$n$ sequence ($n$ units of 20 equally separated $\pi$-pulses) sandwiched between two $\pi/2$-pulses and a waiting time. The numbers associated with the microwave pulses indicate the angle between the rotation axes and the $y$-axis

The control scheme for one measurement cycle is shown in Fig. 1e. The electron spin is optically pumped into the state $|0\rangle \equiv |m_S = 0\rangle$ and then rotated to the $x$-axis state $|+x\rangle$ by a $\pi/2$ pulse about the $y$-axis. The $y$ component of the electron spin is measured by applying a $(-\pi/2)$ pulse around the $x$-axis followed by a projective measurement along the $z$-axis via optical excitation and fluorescence detection. The correlation function is extracted from the photon count statistics (see Methods and Supplementary Note 5). Between the initialization and readout of each cycle, we apply a sequence of $N_p$ equally spaced $\pi$-pulses, i.e., a dynamical decoupling (DD) control on the electron spin to modulate the hyperfine interaction for a total duration of $t_I = N_p\tau$ (with $\tau$ being the pulse interval). The Knill DD sequence KDD$n$, which contains $n$ units of 20 pulses applied along different axes (Fig. 1e) is chosen, to tolerate pulse errors in the many-pulse DD control[33].

The evolution during the DD can be factorized into a control-phase gate and a free precession of the nuclear spin $\hat{U} = \exp\left(-i\Phi\hat{I}_Z\right)\exp\left(2i\alpha\hat{S}_Z\hat{I}_X\right)$ (see Supplementary Note 2). If the hyperfine interaction is weak, the precession angle $\Phi \cong 2\pi\left|\gamma_{13_C}\mathbf{B} - \mathbf{A}/2\right|t_I \equiv 2\pi\bar{\nu}t_I$ (with the $^{13}$C gyromagnetic ratio

$\gamma_{13_C} = -1{,}070.5\,\text{Hz}\,(\text{Gs})^{-1}$ and $\bar{\nu}$ denoting the hyperfine-renormalized Larmor frequency) and the conditional phase shift $\alpha \cong \frac{2A_\perp}{\bar{\nu}}\left|\frac{\sin\left(N_p\pi\bar{\nu}\tau\right)}{\cos(\pi\bar{\nu}\tau)}\right|\sin^2\frac{\pi\bar{\nu}\tau}{2}$ (with $A_\perp$ denoting hyperfine interaction in the $X$–$Y$ plane)[9,23–25,34].

The conditional phase shift $\alpha$ and hence the measurement strength is controllable by changing the DD timing and length. In particular, if the resonant DD condition $2\tau \approx \frac{1}{\bar{\nu}}$ is satisfied, $\alpha \cong 2N_pA_\perp\tau = 2A_\perp t_I$, which is proportional to the number of DD pulses[9,23–25,34].

The free precession angle per measurement cycle $\Phi$ is also controllable by inserting a waiting time between DD control of neighboring cycles. For resonant DD, $\Phi \approx 0 \bmod(2\pi)$. If the magnetic field is much stronger than the hyperfine interaction, i.e., $\gamma_{13_C}B \gg A$, which is the case in our experiment since the former is ~MHz and the latter ~kHz, the $Z$-axis of the free precession is nearly the same as the $z$-axis (the magnetic field direction). The free precession angle per measurement cycle then becomes $\Phi = 2\pi\bar{\nu}t_c \bmod 2\pi$, where $t_c$ is the duration of a measurement cycle including the DD duration $t_I$, the readout and initialization time and the waiting time.

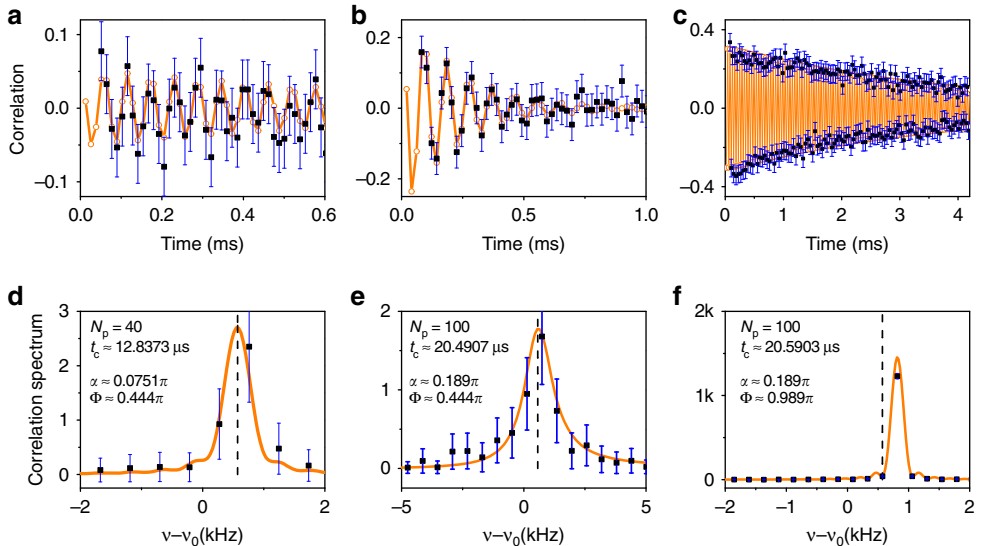

**Fig. 2** Control of the measurement strength relative to the precession frequency of a nuclear spin. **a–c** The correlation functions of the sequential weak measurements on the nuclear spin as functions of the delay time ($Nt_c$) between two measurements separated by $N$ cycles, with $t_c$ being the duration of each measurement cycle. The dark blue error bars represent two standard deviations ($2\sigma$). **d–f** The corresponding correlation spectra. The experimental data are shown in black squares, and theoretical results are in orange lines and open circles. The interval between dynamical decoupling (DD) pulses, $\tau = 0.18224$ μs, is set near the resonance of the Larmor frequency $\nu_0 = 2.743189$ MHz. In (**a/d**), (**b/e**) and (**c/f**), correspondingly, the number of DD pulses is $N_p = 40$, 100, and 100, the duration of one measurement cycle is $t_c = 12.8373$, 20.4907 and 20.5903 μs. The theoretical results reproduce the experimental data well, with the hyperfine coupling parameters $A_Z = 1.142$ kHz and $A_\perp = 16$ kHz. This yields the precession angle $\Phi = 0.445\pi$, $0.443\pi$ and $0.990\pi$, as well as the conditional phase shift $\alpha = 0.0743\pi$, $0.186\pi$ and $0.186\pi$, correspondingly in (**a/d**), (**b/e**) and (**c/f**). The vertical dashed lines in (**d**), (**e**) and (**f**) mark the positions of the hyperfine-modified precession frequency $\bar{\nu} - \nu_0$. All error bars were calculated as explained in Supplementary Note 5. Source data are provided as a Source Data file

Figure 2 shows the control of the measurement strength relative to the precession rate of a $^{13}$C spin weakly coupled to an NV center. The nuclear spin Larmor frequency $\nu_0 = \left|\gamma_{13_C}B\right| \approx 2.743189$ MHz. The interval between neighboring DD pulses is set to satisfy the near-resonance condition ($2\nu_0\tau \approx 1.0001$). The oscillation frequency of the correlation function $\nu_{eff}$ is determined relative to the bare Larmor frequency $\nu_0$. The experimental data are reproduced well in theory with the hyperfine interaction parameters $A_Z = 1.142$ kHz and $A_\perp = 16$ kHz. In Fig. 2a the number of DD pulses is $N_p = 40$. The measurement strength is chosen small compared to the free precession ($\alpha = 0.0743\pi \ll \Phi \approx 0.445\pi$). The correlation function oscillates coherently with slow decay (Fig. 2a), and the spectral peak is close to the Larmor frequency $\bar{\nu} \approx \nu_0 + 0.571$ kHz, with a small broadening due to the measurement-induced dephasing (Fig. 2d). By increasing the number of pulses ($N_p = 100$), and hence the measurement strength ($\alpha = 0.189\pi$), while keeping the free precession angle per measurement cycle $\Phi$ nearly invariant, the decay of the oscillating correlation function (Fig. 2b) and the spectral broadening (Fig. 2e) become more significant. Making the precession angle $\Phi$ close to $\pi$ by adjusting the cycle duration (the theoretical value of $\Phi$ is about $0.990\pi$), while keeping the measurement strength the same ($\alpha = 0.189\pi$), we observe the correlation function to decay exponentially with alternating sign (Fig. 2c). The spectral peak is pinned at $\frac{1}{2t_c}$, about 246 Hz relative to $\bar{\nu} \mod (1/t_c)$ (see Fig. 2f), which indicates that the nuclear spin dynamics is coherently trapped.

The phase transition in quantum dynamics between coherent oscillation and coherent trapping of the nuclear spin can be seen in Fig. 3. The phase boundary $\mu^2 \equiv \tan^4\left(\frac{\alpha}{2}\right) = \sin^2\Phi$ is indicated by the black lines in Fig. 3a, b, which present the dependence of: (a) the effective angular frequency $\Phi_{eff} = 2\pi\nu_{eff}t_c$ and (b) the

decay per measurement cycle $\gamma_{eff}$ of the correlation function on the measurement strength (in terms of $\alpha$) and the nuclear spin free precession frequency (in terms of $\Phi = 2\pi\bar{\nu}t_c$). For relatively weak measurements ($\mu^2 < \sin^2\Phi$), the nuclear spin performs coherent oscillations; otherwise, the nuclear spin is coherently trapped along one direction (for $\cos\Phi > 0$) or alternating in opposite directions (for $\cos\Phi < 0$).

In the experiment, by fixing the number of DD pulses (and hence the measurement strength $\alpha$), the phase transition is observed by varying the measurement cycle duration $t_c$ (hence $\Phi$). Examples are shown in Fig. 3c, d for two different measurement strengths (corresponding to the short horizontal lines in Fig. 3a, b). The effective frequencies and decay rates of the correlation function agree well with the theoretical predictions (with two fitting parameters $A_Z$ and $A_\perp$ the same as in Fig. 2). The transition between frequency dragging and trapping can be seen clearly, and so is a sudden change in the derivative of the decay rate.

**Enhancing the spectral resolution by mitigating back-action.** The measurement-induced decay (and resonance broadening) can be made arbitrary small by choosing an arbitrarily small measurement strength. In the weak measurement limit ($\alpha \to 0$ and $\mu^2 \ll \sin^2\Phi$), the frequency dragging is negligible, $\nu_{eff} \cong \bar{\nu} + \bar{\nu}\frac{\mu^2}{(2\sin^2\Phi)}$, and the spectral resolution, limited by the measurement-induced broadening, is $\delta\nu = \frac{\gamma_{eff}}{(2\pi t_c)} \cong \frac{\alpha^2}{(8\pi t_c)}$.

To demonstrate spectral resolution beyond the $1/T_1$ limit of the sensor electron spin, we choose an NV center (referred to as NV2) in the same diamond crystal as used for Figs. 2, 3 but with weaker coupled $^{13}$C nuclear spins. To enhance the photon count contrast between different states of the NV center spin, we perform repetitive readout of the electron spin for 40 times

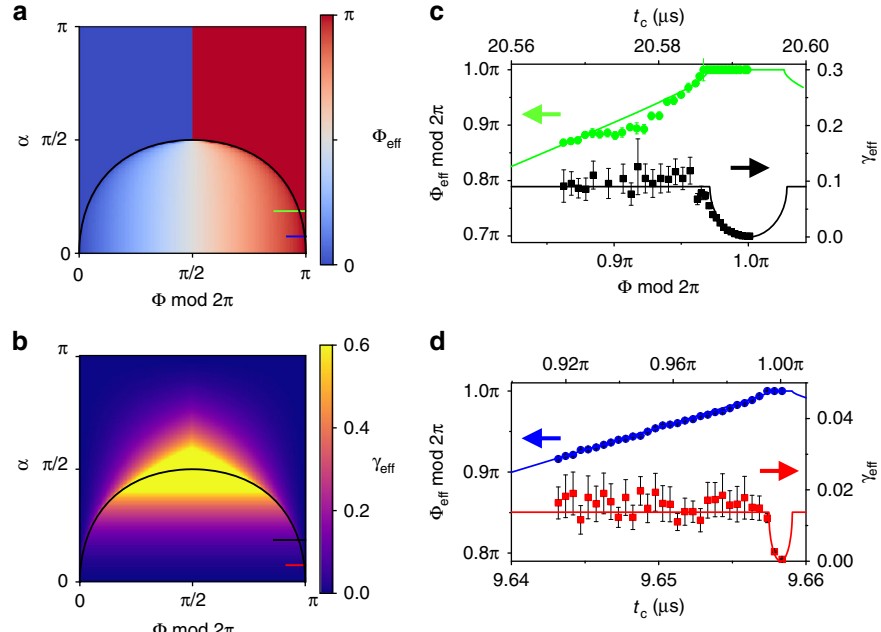

**Fig. 3** Quantum dynamics phase transition between oscillating and trapped dynamics. The theoretical results of **a** the effective precession angle per cycle $\Phi_{eff} = 2\pi\nu_{eff}t_c$ and **b** the effective decay per cycle $\gamma_{eff}$ of the correlation functions versus the conditional phase shift $\alpha$ and nuclear precession angle per measurement cycle $\Phi = 2\pi\bar{\nu}t_c$. The black curves show the phase boundary $\tan^4\left(\frac{\alpha}{2}\right) = \sin^2\Phi$. The short horizontal lines indicate the parameter ranges of the experimental data in (**c**) and (**d**). **c** The experimental oscillation frequency (green circles) and decay per cycle (black squares) of the correlation function as functions of the measurement cycle duration $t_c$ for KDD5 control ($N_p = 100$), compared with the theoretical results (curves) as functions of the nuclear spin precession angle per cycle $\Phi = 2\pi\bar{\nu}t_c$ for $\alpha = 0.186\pi$, **d** The same as (**c**) but for KDD2 control in the experiment (blue circles denote the experimental oscillation frequency, red squares the decay per cycle). The number of pulses is $N_p = 40$, corresponding to $\alpha = 0.0743\pi$ in theory). The nuclear spin, the dynamical decoupling (DD) sequences and the fitting parameters $A_Z = 1.142$ kHz and $A_\perp = 16$ kHz for determining $\Phi$ and $\alpha$ in theory are the same as in Fig. 2. All error bars were calculated as explained in Supplementary Note 5. Source data are provided as a Source Data file

assisted by the $^{14}$N nucleus[35,36] during the waiting time (see Methods and Supplementary Note 1).

The correlation spectrum for $N_p = 100$, as shown in Fig. 4a, exhibits a narrow peak, with half width at half maximum (HWHM) of 1.9 Hz (C1), and a broader one (C2), with HWHM of 8.75 Hz, both well below the $1/T_1$ limit (about 80 Hz) of the NV center electron spin (which has $T_1 \approx 2$ ms). The two resonances are ascribed to two $^{13}$C nuclear spins, with longitudinal hyperfine coupling $A_Z = 19 \pm 2$ Hz (C1) and $A_Z = 178 \pm 8$ Hz (C2). By increasing the number of DD pulses to $N_p = 300$ and hence enhancing the measurement strength (Fig. 4b), the two resonances are broadened. The dependence of the HWHM, as shown in Fig. 4c, agrees well with the measurement-induced broadening plus an additional broadening $\Gamma_0$,

$$\delta\nu = \frac{\alpha^2}{8\pi t_c} + \Gamma_0 = \frac{A_\perp^2 t_I^2}{2\pi t_c} + \Gamma_0. \tag{3}$$

The transverse hyperfine coupling parameter $A_\perp$ is fitted to be about 2.2 kHz (C1) and 4.05 kHz (C2), and the additional broadening $\Gamma_0$ is fitted to be about 0.29 Hz (C1) and 6.24 Hz (C2). In addition to intrinsic broadening (due to, e.g., dipolar interaction with other nuclear spins), one possible contribution to the additional broadening $\Gamma_0$ is the hyperfine coupling fluctuation due to the NV center electron jumping randomly among different levels during readout (details in Supplementary Note 4). It is estimated to be $\Gamma_0^{hf} \approx \frac{A_Z^2 \tau_{eff}^2}{(2\pi t_c)}$, with $\tau_{eff}$ denoting the effective period in each cycle, during which the NV center state is stays in a random state ($\tau_{eff} \sim 600\,\mu s$ in the experiment of Fig. 4). $\Gamma_0$ of C2 is about 30 times greater than that of C1, which is consistent with the larger $A_Z$ of C2. From Fig. 4c it is clear that

the spectral resolution in all the cases studied is limited by the measurement-induced dephasing.

A key aspect of the weak measurement protocol is the data acquisition time $T^D$ for achieving a given spectral resolution $\Delta\nu$ (see Supplementary Note 6)

$$T^D \propto \frac{1}{\Delta\nu}. \tag{4}$$

For comparison, to achieve a resolution beyond the limit set by the lifetime of the sensor, a Ramsey-like scheme is also possible, in which the target spin undergoes an initialization–precession–measurement process in each readout step. The precession time between the initialization and the measurement determines the spectral resolution. However, for a weakly coupled nuclear spin with hyperfine coupling strength $A_\perp < \frac{1}{(2T_1)}$ ($T_1$ denoting the sensor spin relaxation time), a regime of interest in this paper, the data acquisition time in the Ramsey scheme is larger than that of the sequential weak measurement scheme by a factor of $\sim\frac{1}{(2A_\perp T_1)^4}$ (see Supplementary Note 6) since no measurements are performed during the free precession time. For example, to resolve a $^{13}$C nuclear spin located 6 nm away from the NV center (which has a hyperfine interaction of about 90 Hz), the data acquisition time of the Ramsey protocol is longer by a factor of about 1000. Further optimization[18] reduces the measurement time of the Ramsey protocol such that the choice of the optimum method depends on the application at hand (see Supplementary Note 6).

## Discussion
The sequential weak measurement can in principle realize arbitrary spectral resolution of NMR by further reducing the

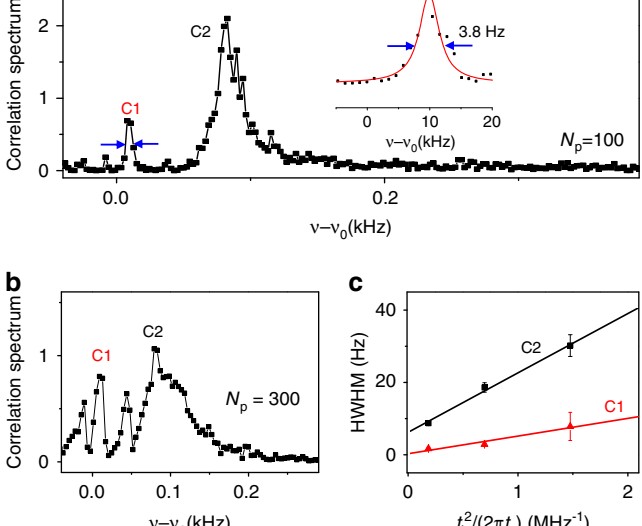

**Fig. 4** High-resolution spectroscopy of single nuclear spins. **a, b** Correlation spectra of sequential measurements of single nuclear spins weakly coupled to the sensor NV2, for the KDD pulse number $N_p = 100$ and 300, respectively. The two peaks C1 and C2 are ascribed to two different nuclear spins. The inset in (**a**) is a close-up of the peak C1, with a Lorentzian fit (red line). **c** The half width at half maximum (HWHM) of the C1 (red triangles) and C2 (black squares) resonances as functions of the interaction time squared (which is proportional to the measurement strength). The symbols are experimental data for $N_P = 100$, 200 and 300, and the curves represent the fitted theory. The linear dependence indicates the broadening is caused mainly by measurement-induced dephasing. The error bars were calculated as explained in Supplementary Note 5. Source data are provided as a Source Data file

measurement strength (via, e.g., choice of DD sequences), and by suppressing the background broadening (via, e.g., the use of more purified crystals and target spins located further away from the sensor—see Supplementary Note 5 for discussions on the spatial range of detection). This method will be particularly useful for NMR of single molecules on diamond surfaces where the hyperfine interaction with the sensor is very weak (e.g., 10 Hz to 0.1 kHz). Since the sensor itself does not limit the resolution, it is also applicable to other solid state spin systems. The sensing efficiency can be enhanced further by employing fast and efficient methods for electron readout[37]. The coherent oscillation and trapping dynamics, tunable by DD control of the central electron spins, can be exploited to control and initialize remote nuclear spins, respectively. Now that schemes are available to spectrally resolve, initialize, and coherently control multiple nuclear spins that are not required to be located closely to a central electron spin, quantum information processing with a relatively large number (e.g., >10) of nuclear spins is a step closer[38].

## Methods

**Weak measurement formalism and correlation function.** The mutually controlled phase gate is $e^{2i\alpha \hat{S}_z \hat{I}_X}$, where the coordinate axes $(x, y, z)$ of the sensor and those of the target $(X, Y, Z)$ are not necessarily identical. The sensor spin is initially in the state $|x\rangle$ and at the end measured along the $y$-axis. The weak measurement is characterized by the Kraus operators $\hat{M}_\pm = \langle \pm y | e^{2i\alpha \hat{S}_z \hat{I}_X} | x \rangle = \frac{(e^{i\alpha \hat{I}_X} + i e^{-i\alpha \hat{I}_X})}{2}$. Given the initial state of the target spin as described by a density matrix $\hat{\rho}$, the probability of the output $m_k = \pm 1$ is $p_\pm = \text{Tr}\left(\hat{M}_\pm \hat{\rho} \hat{M}_\pm^\dagger\right) = \frac{1}{2} \pm I_X \sin\alpha$ with $I_X = \text{Tr}(\hat{\rho} \hat{I}_X)$

and the state after the measurement is $\hat{\rho}_\pm = \frac{(\hat{M}_\pm \hat{\rho} \hat{M}_\pm^\dagger)}{p_\pm}$. In particular for a fully unpolarized initial state $\hat{\rho} = 1/2$, the post-measurement state is $\hat{\rho}_\pm = \frac{1}{2} \pm \hat{I}_X \sin\alpha$ corresponding to the output $m_k = \pm 1$, with partial polarization $\pm \sin\alpha$ along the $X$-axis, which is called heralded initialization. If the output is discarded, the target state becomes

$$\hat{\mathcal{M}}[\hat{\rho}] \equiv \hat{M}_+ \hat{\rho} \hat{M}_+^\dagger + \hat{M}_- \hat{\rho} \hat{M}_-^\dagger = \hat{\rho}\cos^2\frac{\alpha}{2} + 4\hat{I}_X \hat{\rho} \hat{I}_X \sin^2\frac{\alpha}{2}, \tag{5}$$

which reduces the spin polarization in the $Y$–$Z$ plane by a factor of $\cos\alpha$, that is, pure dephasing in the $X$ basis (which reduces the spin polarization along the $Y$- and $Z$-axes, but keeps the $X$ component unchanged).

After the measurement, the target spin can undergo a free precession, e.g., about the $Z$-axis via the evolution $\hat{\mathcal{U}}[\hat{\rho}] = e^{-i\Phi \hat{I}_Z} \hat{\rho} e^{i\Phi \hat{I}_Z}$. The correlation function is

$$C(N) \equiv \langle m_{k+N} m_k \rangle = \text{Tr}\left[\hat{\mathcal{P}}\left(\hat{\mathcal{U}}\hat{\mathcal{M}}\right)^{N-1} \hat{\mathcal{U}}\hat{\mathcal{P}}[\hat{\rho}]\right], \tag{6}$$

where the polarization operator $\hat{\mathcal{P}}[\hat{\rho}] \equiv \hat{M}_+ \hat{\rho} \hat{M}_+^\dagger - \hat{M}_- \hat{\rho} \hat{M}_-^\dagger$ denotes the heralded initialization.

The precession of the target spin polarization **I** is described by the transform

$$\mathcal{U}\begin{pmatrix} I_X \\ I_Y \\ I_Z \end{pmatrix} = \begin{pmatrix} \cos\Phi & -\sin\Phi & 0 \\ \sin\Phi & \cos\Phi & 0 \\ 0 & 0 & 1 \end{pmatrix}\begin{pmatrix} I_X \\ I_Y \\ I_Z \end{pmatrix}. \tag{7}$$

The measurement transforms the polarization by

$$\mathcal{M}\begin{pmatrix} I_X \\ I_Y \\ I_Z \end{pmatrix} = \begin{pmatrix} 1 & 0 & 0 \\ 0 & \cos\alpha & 0 \\ 0 & 0 & \cos\alpha \end{pmatrix}\begin{pmatrix} I_X \\ I_Y \\ I_Z \end{pmatrix}. \tag{8}$$

The eigenvalues of the transformation $\mathcal{UM}$ are easily obtained as $\eta_Z = \cos\alpha$ and $\eta_\pm = \left(\cos\Phi \pm \sqrt{\mu^2 - \sin^2\Phi}\right)\cos^2\frac{\alpha}{2}$ with $\mu \equiv \tan^2\left(\frac{\alpha}{2}\right)$, corresponding to the right eigenvectors $\mathbf{v}_Z^R = (0, 0, 1)^T$ and $\mathbf{v}_\pm^R = \left(\cos\Phi\sin^2\frac{\alpha}{2} \pm \Delta, \sin\Phi, 0\right)^T$ with $\Delta = \cos^2\frac{\alpha}{2}\sqrt{\mu^2 - \sin^2\Phi}$, and left eigenvectors $\mathbf{v}_Z^L = (0, 0, 1)$ and $\mathbf{v}_\pm^L = \frac{1}{2\Delta\sin\Phi}\left(\pm\sin\Phi, \mp\cos\Phi\sin^2\frac{\alpha}{2} - \Delta, 0\right)$. They satisfy the orthonormal conditions $\mathbf{v}_i^L \mathbf{v}_j^R = \delta_{ij}$ and $\mathbf{v}_+^R \mathbf{v}_+^L + \mathbf{v}_-^R \mathbf{v}_-^L + \mathbf{v}_Z^R \mathbf{v}_Z^L = 1$. For the target spin initially in the fully unpolarized state, the heralded initialization polarizes the spin along the $X$-axis to be $(\sin\alpha, 0, 0)^T = \sin\alpha\mathbf{e}_X$ and the following measurement and precession keeps it in the $X$–$Y$ plane. So only the last two eigenstates of $\mathcal{UM}$ are relevant. The correlation function is

$$C(N) = \sin^2\alpha\left[\left(\mathbf{e}_X^T\mathbf{v}_+^R\mathbf{v}_+^L\mathbf{e}_X\right)\eta_+^N + \left(\mathbf{e}_X^T\mathbf{v}_-^R\mathbf{v}_-^L\mathbf{e}_X\right)\eta_-^N\right]$$
$$= \sin^2\alpha\left(\frac{\eta_+^N + \eta_-^N}{2} + \frac{\eta_+^N - \eta_-^N}{\eta_+ - \eta_-}\cos\Phi\sin^2\frac{\alpha}{2}\right). \tag{9}$$

1. If $\mu^2 < \sin^2\Phi$, the correlation function oscillates with a dragged frequency and an effective decay;

2. If $\mu^2 > \sin^2\Phi$ and $\cos\Phi > 0$, the correlation function for large $N$ is dominated by $\eta_+^N$, an exponential decay corresponding to trapping along the axis rotated from the $X$-axis by an angle $\approx\Phi$ about the $Z$-axis;

3. For $\mu^2 > \sin^2\Phi$ and $\cos\Phi < 0$, the correlation function for large $N$ is dominated by $\eta_-^N$, an exponential decay with alternating sign $(-1)^N$ corresponding to coherent trapping alternatively parallel and anti-parallel to the axis rotated from the $X$-axis by an angle $\approx\Phi$ about the $Z$-axis.

The correlation function for multiple nuclear spins is discussed in Supplementary Note 3.

**Diamond crystal.** A 99.995% $^{12}$C-enriched diamond crystal (5.3 mm × 4.7 mm × 2.6 mm) was grown by the temperature gradient method under high-pressure high-temperature conditions of 5.5 GPa and 1350 °C using high-purity Fe–Co–Ti solvent and high-purity $^{12}$C-enriched solid carbon. The crystal was irradiated by 2 MeV electrons at room temperature to the total fluence of $1.3 \times 10^{11}$ cm$^{-2}$ and annealed at 1000 °C (for 2 h in vacuum) to create single NV centers from intrinsic nitrogen impurities. A polished, (111)-oriented slice (2 mm × 2 mm × 80 μm) obtained by laser-cutting has been used in the present work. The isotopic enrichment enables the detection of single, weakly coupled $^{13}$C nuclear spins, and mitigates the existence of a strong, overlapping $^{13}$C spin bath. The $T_2^*$ is typically on the order of 50 μs.

**Experimental setup.** The diamond crystal is positioned inside a superconducting vector magnet ($B_z = 3$ T, $B_{x,y} = 0.2$ T), with the diamond surface normal pointing

along the main magnetic field axis. The magnet is running at a field of about $B = 2561$ Gs with a magnetic field stability of $\sim 2.5 \cdot 10^{-5}$ Gs h$^{-1}$ (corresponding to $^{13}$C Larmor frequency stability $\sim 0.0274$Hz h$^{-1}$). For a typical accumulation time of $\sim 3$ h for one measurement, this results in a negligible drift of the $^{13}$C Larmor frequency of about 0.08 Hz. The $^{13}$C Larmor frequency $\nu_0 = |\gamma_{^{13}C}B|$ is measured to be $2{,}743{,}189 \pm 190$ Hz for Figs. 2, 3 and $2{,}740{,}090.4 \pm 7.9$ Hz for Fig. 4 (see Supplementary Note 5 and Supplementary Figure 6). The magnetic field shifts the NV center transition $|m_S = 0\rangle \leftrightarrow |m_S = -1\rangle$ to around 4.3 GHz.

The experiment consists of a home-built confocal microscope with a 520 nm excitation laser diode. The laser can be switched on and off on the timescale of 10 ns. The photoluminescence of single NV centers is collected via an oil-immersion objective with a numeric aperture of 1.35 and detected by an avalanche photodiode, capable of detecting single photons. The spin resonance is detected optically via spin state dependent fluorescence of single NV centers. Microwave radiation is generated by an arbitrary waveform generator (AWG) with a sampling rate of 12 Gigasample s$^{-1}$, and subsequent amplification up to a power of around 40 dBm. The same AWG also controls the timing of the experiment. The microwaves are guided through coaxial cables and a coplanar waveguide, with a width of around 100 μm at the position of the NV. The radio frequency signal used to manipulate the nitrogen nuclear spin is directed through the same waveguide.

**Nuclear spin assisted readout**. One single readout of the NV center spin, via a 300 ns laser pulse, produces much less than one photon on average. The readout efficiency can be increased, by transferring the spin state from the NV center electron spin to the NV center $^{14}$N nuclear spin, which can subsequently be read out multiple times[35,36]. Since only the $m_S = 0$ and $m_S = -1$ spin levels of the NV center spin are used for sensing, the $^{14}$N nuclear spin consequently has to be initialized into a two-level manifold (in this work $m_I = 0$ and $m_I = +1$) before every measurement. For more information, see Supplementary Note 1.

**Construction of the correlation function from photon statistics**. Photon counts in experiments are not a perfect measurement of the NV center spin. We use $D(n_k|m_k)$ to denote the probability of detection of $n_k$ photons for the NV center spin state that would yield an output $m_k$ in a perfect measurement. The joint probability of detection of $n_k$ and $n_{k+N}$ for the $k$-th and $(k+N)$-th measurements is

$$p(n_k, n_{k+N}) = \sum_{m_k, m_{k+N}} D(n_k|m_k) D(n_{k+N}|m_{k+N}) p(m_k, m_{k+N}), \quad (10)$$

where the joint probability $p(m_k, m_{k+N}) = \frac{\left[1 + m_k m_{k+N} C(N)\right]}{4}$ depends on the correlation function. Direct calculation yields

$$C(N) = \frac{4\left(\langle n_k n_{k+N}\rangle - \bar{n}^2\right)}{\left(\bar{n}_+ - \bar{n}_-\right)^2}, \quad (11)$$

where $\bar{n}_\pm$ is the averaged photon count for the senor measurement output $\pm 1$ and $\bar{n} \equiv \frac{\left(\bar{n}_+ + \bar{n}_-\right)}{2}$ is the average photon count.

**Code availability**. Custom computer code used in the theoretical studies and experimental analysis are available from the corresponding authors upon reasonable request

## Data availability

Data supporting the findings of this study are available within the article and its Supplementary Information and from the corresponding authors upon request. The source data underlying Figs. 2a–f, 3a–d, 4a–c, Supplementary Figures 2, 3a, b, 4a–c, 5, 6, 7a–c, 8a–f and 9a–c are provided as a Source Data File.

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

## Acknowledgements

We thank S. Zaiser and N. Aslam for fruitful discussions. We acknowledge financial support by the German Science Foundation (SPP1601, FOR2724), the EU (ASTERIQS, SMel), the Max Planck Society, the Volkswagen Stiftung, the Hong Kong Research Grants Council General Research Fund (No. 14319016), the MOST of China (Grants No. 2014CB848700) and the Japan Science and Technology Agency and Japan Society for the Promotion of Science KAKENHI (Nos. 26246001 and 26220903).

## Author contributions

R.-B.L. proposed and supervised the theoretical study; R.-B.L., W.Y., P.W. and X.Y.P. formulated the theory; P.W. and W.Y. carried out the calculations; M.P., P.N. and J.W. conceived the experiments; M.P. carried out the measurements; H.S., S.O. and J.I. designed and conducted the synthesis and fabrication of the diamond substrate; M.P., P.W., P.N., W.Y., R.-B.L. and J.W. analyzed the data; P.W., M.P., D.B.R.D., W.Y., R.B.L. and J.W. wrote the manuscript with contributions from other authors.

## Additional information

**Competing interests:** The authors declare no competing interests.

