## [Peer Review File · Nature Communications]

Reviewers' comments:

Reviewer #1 (Remarks to the Author):

The manuscript by Pfender et al. reports the realization of sequential weak measurements on a single spin-half system. The authors use a probe system (the spin of an electron) to make weak measurements on the nuclear spin, allowing them to measure the spin precession frequency to unprecedented accuracy. The authors also explore the dynamical effect of the weak measurements on the spin dynamics. Since this work is a significant step in realizing more sophisticated measurement protocols that may lead to practical high-precision measurement techniques, and is also a realization of a paradigmatic quantum measurement that plays a central role in the theoretical understanding of quantum measurement and feedback processes, I think the manuscript clearly crosses the threshold for publication in Nature Communications.

Reviewer #2 (Remarks to the Author):

The manuscript from Pfender, et al. reports on and addresses the issue of the 'back action' imparted by quantum sensors on their target system. These effects are particularly problematic where such sensors are employed as NMR spectrometers of small number of nuclear spins, as it leads to an artificial degradation of spectral resolution. Via a detailed analytical treatment, the authors develop and demonstrate a measurement strategy whereby the effects of this back-action may be arbitrarily suppressed via interleaved periods of precisely controlled continuous measurement and free evolution of the target nuclei.

The basic measurement scheme is introduced in general terms, and subsequent development and demonstration of the method are carried out in the context of the electronic spin of a nitrogen vacancy defect centre in diamond as the sensor qubit. The authors discuss how the parameters associated with successive, synchronised Knill dynamical decoupling (KDDn) pulse sequences may be tuned to control the contributions to the nuclear spin evolution due to periods of free precession, and that which is conditional upon the resulting longitudinal dynamics of the NV spin.

The correlation function associated with the projection of the nuclear spin along the measurement axis is experimentally measured under three limiting regimes of measurement strength and free precession angles. These results are interpreted in the context of a quantum phase transition between coherent dynamics and coherent trapping of the nuclear spin state. These studies are extremely elegant and the authors should be congratulated on the creative conception and execution of this work.

Perhaps of most interest to the broader quantum sensing community is the final study demonstrating how the measurement induced component of the NMR linewidth may be arbitrarily mitigated via control of the effective measurement strength. Using an NV with two weakly coupled ^{13}C nuclei, the authors show that the linewidth may be reduced to around a few Hz, which is well below the longitudinal relaxation rate of the NV. The measurement strength is controllably increased from 100 pulses to 300, and the linear dependence on this parameter subsequently exhibited by the spectral linewidth shows that additional dephasing is in fact induced by increasing measurement strength.

This work addresses a critical outstanding problem in the field of NV-based NMR; that being the trade-off between spectral resolution and sensitivity. I believe that this development is exciting, the results both valid and compelling, and the surrounding analysis is sufficiently elegant to warrant publication in Nature Communications.

Reviewer #3 (Remarks to the Author):

This is an excellent paper reporting a spectacular set of experiments. The measurement method demonstrated provides a powerful practical technique to measure weakly coupled spins (although I am not convinced it is fundamentally better than other approaches, see below). Additionally, the theoretical analysis is clear and the work certainly provides new insight in the problem of measuring nuclear spectra with a single electron. I also want to highlight that the stability and control of the experimental apparatus that make such precise measurements at room temperature possible are spectacular.

I recommend publication in Nature Communications, given that the authors address the following comment:

I am not convinced that the comparison in supplementary information VI. "Data acquisition time" is fair. For the Ramsey scheme the following sequence is considered:

Measurement/initialization - waiting time - measurement

My problem lies in the choice of the implementation for the final measurement. The authors take the measurement to consist of a single DD sequence that correlates the electron spin with the nuclear spin state and then a readout (measurement) of the electron spin. After that the entire sequence is restarted.

The authors correctly argue that, if the electron coherence or relaxation (T_1) is short compared to the coupling, then the optimal choice for the chosen sequence is a relatively short DD sequence and the measurement of the target spin state is quite weak (poor). If one then restarts the entire sequence, the total time needed indeed scales poorly with T_1 .

However, in the regime analysed by the authors, this measurement implementation is not an optimal choice. The final measurement would consist of several repeated DD + electron readout sequences to build up a strong and high-fidelity readout of the target spin.

The previously made assumptions that (1) decoherence of the target spin other than due to measurement back action can be neglected and (2) that the DD + electron-readout time is negligible (doesn't appear in the equations), guarantees that one can do this at no extra time cost. In other words, under the assumptions made it is possible to perform a (near-)perfect strong projective measurement of the target spin independent of the T_1 of the electron spin. Or, in yet other words, as the measurement does not disturb the target spin beyond the fundamental back-action, it can be probed repeatedly and measured in a single-shot.

Mathematically this means that the factor $\sin(\alpha_R)^2$ in line 509 would be always equal to 1 and in line 532 the result is always the top line. Thus, the Ramsey sequence under these assumptions is actually a factor 2 better.

Note that I do not question the correctness of the calculations in section VI. I just don't agree with the comparison made. One should not repeat the entire Ramsey sequence after a single (weak) probing of the target spin: the very nature of the assumptions that were made in the derivation ensure that the information is still there and that more repetitions will improve the result.

Also note that I do not question the practical power of the weak measurement scheme implemented. It beautifully combines efficient data taking while overcoming the dephasing of the

target spin due to the electron spin relaxation (from a practical perspective it might very well be an optimal choice!). However, I do question the apparent claim based on section VI that there is a fundamental advantage over the basic Ramsey sequence with a strong measurement at the end.

Response to reviewer comments on „**High-resolution spectroscopy of single nuclear spins via sequential weak measurements**”

Matthias Pfender, Ping Wang, Hitoshi Sumiya, Shinobu Onoda, Wen Yang, Durga Bhaktavatsala Rao Dasari, Philipp Neumann, Xin-Yu Pan, Junichi Isoya, Ren-Bao Liu, J. Wrachtrup

October 31st, 2018

We thank the reviewers for reading and evaluating our manuscript. While the comments from reviewer #1 and #2 were consistently positive without concern, reviewer #3 found an inconsistency in our calculations of the measurement accumulation time performed in the supplementary information. We added a paragraph in the supplementary information, to account for the case described by the reviewer, and mention it in the main text.

We furthermore shortened the abstract to 150 words.

Changes to the manuscript and supplementary information are highlighted in yellow.

Comments from reviewer #3

This is an excellent paper reporting a spectacular set of experiments. The measurement method demonstrated provides a powerful practical technique to measure weakly coupled spins (although I am not convinced it is fundamentally better than other approaches, see below). Additionally, the theoretical analysis is clear and the work certainly provides new insight in the problem of measuring nuclear spectra with a single electron. I also want to highlight that the stability and control of the experimental apparatus that make such precise measurements at room temperature possible are spectacular.

I recommend publication in Nature Communications, given that the authors address the following comment:

I am not convinced that the comparison in supplementary information VI. "Data acquisition time" is fair. For the Ramsey scheme the following sequence is considered:

Measurement/initialization - waiting time - measurement

My problem lies in the choice of the implementation for the final measurement. The authors take the measurement to consists of a single DD sequence that correlates the electron spin with the nuclear spin state and then a readout (measurement) of the electron spin. After that the entire sequence is restarted.

The authors correctly argue that, if the electron coherence or relaxation (T_1) is short compared to the coupling, then the optimal choice for the chosen sequence is a relatively

short DD sequence and the measurement of the target spin state is quite weak (poor). If one then restarts the entire sequence, the total time needed indeed scales poorly with T_1 .

However, in the regime analysed by the authors, this measurement implementation is not an optimal choice. The final measurement would consist of several repeated DD + electron readout sequences to build up a strong and high-fidelity readout of the target spin.

The previously made assumptions that (1) decoherence of the target spin other than due to measurement back action can be neglected and (2) that the DD + electron-readout time is negligible (doesn't appear in the equations), guarantees that one can do this at no extra time cost. In other words, under the assumptions made it is possible to perform a (near-)perfect strong projective measurement of the target spin independent of the T_1 of the electron spin. Or, in yet other words, as the measurement does not disturb the target spin beyond the fundamental back-action, it can be probed repeatedly and measured in a single-shot.

Mathematically this means that the factor $\sin(\alpha_R)^2$ in line 509 would be always equal to 1 and in line 532 the result is always the top line. Thus, the Ramsey sequence under these assumptions is actually a factor 2 better.

Note that I do not question the correctness of the calculations in section VI. I just don't agree with the comparison made. One should not repeat the entire Ramsey sequence after a single (weak) probing of the target spin: the very nature of the assumptions that were made in the derivation ensure that the information is still there and that more repetitions will improve the result.

Also note that I do not question the practical power of the weak measurement scheme implemented. It beautifully combines efficient data taking while overcoming the dephasing of the target spin due to the electron spin relaxation (from a practical perspective it might very well be an optimal choice!). However, I do question the apparent claim based on section VI that there is a fundamental advantage over the basic Ramsey sequence with a strong measurement at the end.

Thanks for the referee's suggestion. As pointed out by the referee, building up a strong measurement by repeating weak measurements is beneficial in the Ramsey scheme (see Gefen, T. et al., Phys. Rev. A 98, 013844 (2018)). However, increasing the signal would also increase the measurement time. Specifically, removing the factor $\sin^2 \alpha$ of the signal would increase the measurement duration time by a factor $1/\sin^4 \alpha$ though the free evolution time is kept. For nanoscale magnetic resonance, the measurement time can't be neglected. When this factor is taken into account, we find the weak measurement protocol exhibits a better performance over the Ramsey protocol by a factor $\approx 1/(A_\perp T_1)^3$. Please see the detail analysis in the supplement.

To comply with the referee suggestion and make the reader aware of the various scenarios at which the different approaches are optimal, we added the following sentence to the main manuscript: "Further optimization reduces the measurement time of the Ramsey protocol

such that the choice of the optimum method depends on the application at hand (see Supplementary Information).”

Further we modified the Supplementary Information and as outlined above.

REVIEWERS' COMMENTS:

Reviewer #3 (Remarks to the Author):

In my opinion, the authors have satisfactorily responded to the queries raised by the reviewers. I recommend this paper for publication in Nature Communications.

A remark that the authors might want to consider: It does seem to me that there is more to say about the optimal signal-to-noise ratios and measurement times for the different schemes. In particular it is not obvious to me why the the "finite time for each shot of the measurement" (line 547 of the supplement) needs to be considered for the section starting at line 539 (supplement) and not for the section starting at line 492 (VI data acquisition time, supplement). To me, it is a surprising result that it makes such a big difference in total time if one starts extracting information during the evolution time or afterwards, as the measurement procedure in both cases is very similar in all other ways. Naively one might expect up to a factor 2, not a factor ~ 785 (or more).

Response to reviewer comments on „**High-resolution spectroscopy of single nuclear spins via sequential weak measurements**”

Matthias Pfender, Ping Wang, Hitoshi Sumiya, Shinobu Onoda, Wen Yang, Durga Bhaktavatsala Rao Dasari, Philipp Neumann, Xin-Yu Pan, Junichi Isoya, Ren-Bao Liu, J. Wrachtrup

December 27th, 2018

We thank the reviewer for again reading and evaluating our manuscript.

Comments from reviewer #3

In my opinion, the authors have satisfactorily responded to the queries raised by the reviewers. I recommend this paper for publication in Nature Communications.

A remark that the authors might want to consider: It does seem to me that there is more to say about the optimal signal-to-noise ratios and measurement times for the different schemes. In particular it is not obvious to me why the the "finite time for each shot of the measurement" (line 547 of the supplement) needs to be considered for the section starting at line 539 (supplement) and not for the section starting at line 492 (VI data acquisition time, supplement).

In line 492 of the SI, we indeed have considered the finite time for each shot of the measurement because the total time is equal to Mt_c . Here t_c is the finite measurement time (which includes the initialization, entanglement and readout). However, for the Ramsey protocol, this was previously neglected. In order to enhance the signal, as previously pointed out by the reviewer, the duration of the measurement can be increased. For very weakly coupled nuclear spins, the duration of this process then can't be neglected.

To me, it is a surprising result that it makes such a big difference in total time if one starts extracting information during the evolution time or afterwards, as the measurement procedure in both cases is very similar in all other ways. Naively one might expect up to a factor 2, not a factor ~ 785 (or more).

For a better understanding, why the weak measurement protocol is faster, let's look at how the correlation signal is taken from the measurement data (see the following figure). We can see that correlation $C(2)$ and $C(3)$ for different time differences can be calculated from the same dataset. This is the key reason why the weak measurement protocol is faster when sensing weakly coupled nuclear spin.

For the Ramsey protocol, the data for one evolution time can't be used to calculate the correlation of a different evolution time. As shown in the following figure, the data of C(1) cannot be used to calculate C(2). For this reason, the Ramsey protocol wastes a lot of time, if the nuclear spin has weak coupling (small measurement strength). Though the signal can be enhanced by combing many measurement cycles for each evolution time t_1 , the time cost is still longer than that of the weak measurement protocol, as pointed in the supplementary information.